# Triboelectrification induced self-powered microbial disinfection using nanowire-enhanced localized electric field

Zheng-Yang Huo[1,6], Young-Jun Kim[1,6], In-Yong Suh[1], Dong-Min Lee[1], Jeong Hwan Lee[1], Ye Du[2], Si Wang[1,3], Hong-Joon Yoon[1] & Sang-Woo Kim[1,4,5 ✉]

Air-transmitted pathogens may cause severe epidemics showing huge threats to public health. Microbial inactivation in the air is essential, whereas the feasibility of existing air disinfection technologies meets challenges including only achieving physical separation but no inactivation, obvious pressure drops, and energy intensiveness. Here we report a rapid disinfection method toward air-transmitted bacteria and viruses using the nanowire-enhanced localized electric field to damage the outer structures of microbes. This air disinfection system is driven by a triboelectric nanogenerator that converts mechanical vibration to electricity effectively and achieves self-powered. Assisted by a rational design for the accelerated charging and trapping of microbes, this air disinfection system promotes microbial transport and achieves high performance: >99.99% microbial inactivation within 0.025 s in a fast airflow (2 m/s) while only causing low pressure drops (<24 Pa). This rapid, self-powered air disinfection method may fill the urgent need for air-transmitted microbial inactivation to protect public health.

[1] School of Advanced Materials Science and Engineering, Sungkyunkwan University (SKKU), Suwon, Republic of Korea. [2] College of Architecture and Environment, Sichuan University, Chengdu, PR China. [3] State Key Laboratory of Electronic Thin Films and Integrated Devices, School of Optoelectronic Science and Engineering, University of Electronic Science and Technology of China (UESTC), Chengdu, PR China. [4] SKKU Advanced Institute of Nanotechnology (SAINT), Sungkyunkwan University (SKKU), Suwon, Republic of Korea. [5] Samsung Advanced Institute for Health Sciences & Technology (SAIHST), Sungkyunkwan University (SKKU), Suwon, Republic of Korea. [6]These authors contributed equally: Zheng-Yang Huo, Young-Jun Kim. ✉email: kimsw1@skku.edu

Air-transmitted pathogens are the primary reason for people catching pneumonia, asthma, and influenza, as such, they remain a huge threat to public health[1]. When a person is infected by the pathogenic microbes, micron-sized aerosols containing the hazardous bacteria/viruses may release into the air by daily motions such as breaths, coughs, or sneezes[2,3]. The parts of the aerosols with a small diameter (<1 μm) can float in the air for long distances (~kilometers) and so significantly increase the chance of infecting surrounding people, especially in indoor environments such as in hospitals, offices, and restaurant as well as on aircraft and cruises[4–6]. However, the most commonly used air-transmitted pathogen removal method, high-efficiency particulate air (HEPA) filtration, causes an obvious pressure drop where it is applied and cannot inactivate the separated pathogens[7]. In addition, with HEPA, during the regular filter replacement process, any enriched pathogens caught in the filter may be released and cause a second wave of airborne contamination[8]. Alternative methods such as UV radiation and photocatalytic disinfection also suffer drawbacks such as low throughput, intensive energy consumption, and/or strong dependence on sunlight[9–13].

Electroporation, a physical process relying on a strong electric field to damage the outer structure of microbes (bacterial membranes and viral capsids), offers more opportunities for microbial disinfection[14–16]. To generate highly localized electric fields, nanowires (e.g., Ag, CuO, and ZnO nanowires) are placed vertically on a flat electrode surface to enhance the field at the wire tip significantly (>$10^7$ V m$^{-1}$) using only low drive voltages (several V)[17–19]. In fact, a nanowire-assisted electroporation disinfection technology has recently been confirmed as feasible for bacterial disinfection in water[20,21]. However, due to the enhanced electric field existing only near the nanowire tip, unless the microbes approach the tip structure, disinfection will not occur. Thus, the microbial transport process is the speed imitating step of nanowire-assisted electroporation disinfection[22]. In addition, the feasibility of deploying electroporation disinfection in the air remains a great challenge due to the fast airflow in the ventilation systems of indoor buildings (~m/s; >200 folds the flow rate seen in water disinfection)[9,23]. Another concern is that the electroporation disinfection shows a strong dependence on an external power supply. Such drawback limits the application of this method in certain buildings and/or in rural areas, where adding external power supplies in the complex ventilation systems is undesirable and/or power is scarce[16].

Triboelectric nanogenerators (TENGs) can harness the kinetic or ambient energy based on the coupling effect of contact-electrification and electrostatic charge induction[24–26]. These devices are lightweight, low cost, and have high voltage output, as such, they are garnering increased interest for application in pathogen disinfection[27]. A self-powered water disinfection system was developed using ZnO nanowire-modified electrodes driven by a ball-in-ball TENG harnessing wave energy to achieve electroporation disinfection of bacteria in water[19]. A hand-powered TENG was also developed to achieve bacterial disinfection in water based on the coupling effect of an enhanced localized electric field and copper ion toxicity[28]. However, all current studies on TENG-driven disinfection methods are commonly feasible in water, whereas the feasibility of such a system for air disinfection is still questionable due to the faster airflow rate (~m/s)[9,29]. In addition, no research has achieved viral disinfection when powered by TENGs.

Here, we report a self-powered disinfection system for the rapid disinfection of air-transmitted bacteria and viruses based on a highly efficient nanowire-assisted electroporation mechanism powered by vibration-driven TENGs (V-TENGs) that harvest mechanical vibration energy. Owing to a rational design for the accelerated charging and trapping of microbes, we were successful in overcoming the speed limitations for nanowire-assisted electroporation disinfection and achieved high air disinfection efficiency. More than 99.99% of bacteria and viruses were inactivated in the air at a fast airflow rate (2 m/s), this corresponds to a treatment time of 0.025 s while maintaining only a low pressure drop (24 Pa). Our work successfully provides a proof-of-concept to confirm the application potential of this method for air disinfection in the ventilation systems of buildings in actual situations.

## Results

**Resonance-vibration-driven (RV) air disinfection methods.** In our work, we aim to provide a proof-of-concept to confirm the feasibility of RV air disinfection methods in the ventilator of buildings during the normal ventilation process. The construction of the RV air disinfection system (Fig. 1a) consists of three components: a V-TENG, a power management system with rectifiers, and a three-electrode disinfection filter for air-transmitted microbial disinfection. A contact-separation-type V-TENG (with top, middle, and bottom layers) was developed and placed on the ventilation systems of a building to convert mechanical vibrations to electricity. To guarantee sufficient power output, the resonance frequency of the V-TENG was designed to be the same as the vibration frequency of the ventilator (~30 Hz) to give a significant vibration amplitude based on the governing equation (Eq. 1):

$$f = \frac{1}{2\pi}\sqrt{\frac{k}{m}} \tag{1}$$

where $f$ is the resonance frequency of the vibration system, $m$ is the mass of the middle layer of the TENG, and $k$ is the spring constant (see design details in Supplementary Note 1)[30,31]. The generated output from the V-TENG is altering current (AC), which can be tuned into direct current (DC) using rectifiers. After rectification, the output will power the three-electrode disinfection filter.

Figure 1b shows illustrations of the disinfection filter in the RV-disinfection system. It is composed of a four-layer stainless-steel macro-mesh (5 mm × 5 mm of square pores) that serves as the negative electrode, a copper-phosphide-nanowire-modified copper plate (Cu₃PNW-Cu) electrode serving as the positive electrode, and a stainless-steel ground electrode integrated with the positive electrode in a parallel structure (Supplementary Fig. 1). The airflow containing microbes (including bacteria or viruses) will flow through the designed duct and passed the negative and the positive/ground electrodes sequentially (see design details in Supplementary Note 2). The process of the RV air disinfection method is explained in the three steps shown in the schematics (Fig. 1b). In Step 1, bacteria and viruses in the air will contact the surface of the macro-mesh electrode to be charged negatively when flowing through the negative electrode. The design of the multi-layer electrode ensures high contact efficiency meanwhile the macro-mesh structure guarantees only a low pressure drop in the airflow. In Step 2, the charged microbes will then flow between the positive and ground electrodes. Owing to the parallel structure of the integrated positive/ground electrodes with a short distance (1 cm) between them, a relatively strong background electric field with a strength of ~100 V/cm exists between the positive and ground electrodes. Thus, when the negatively charged microbes flow through, they can be trapped on the positive electrode surface immediately by electrostatic attraction. The surface of the positive electrode is modified by vertically grown Cu₃PNWs that can generate an enhanced localized electric field (>$10^7$ V/m) near the nanowire tip when

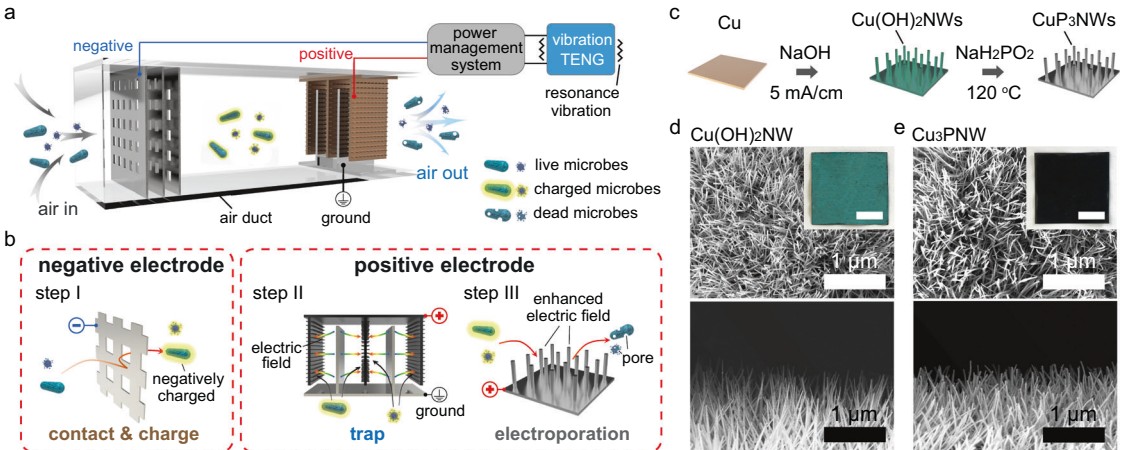

**Fig. 1 Working principle of the resonance-vibration-driven (RV)-disinfection system for air-transmitted microbes. a** Schematics of the RV-disinfection system in an air duct. It consists of a triboelectric nanogenerator (TENG), a power management system with rectifiers, and a three-electrode disinfection filter for air-transmitted microbial disinfection. The resonance frequency of the TENG was designed to be the same as the vibration frequency of the ventilators found in buildings to achieve a significant amplitude for greater power output. **b** Illustrations of the disinfection filter in the RV-disinfection system. It consists of a macro-mesh negative electrode and integrated positive/ground electrodes. In Step 1, microbes (bacteria and viruses) in the air come into contact with the surface of the macro-mesh electrode and are charged negatively when flowing through the negative electrode. In Step 2, the charged microbes then flow between the positive and ground electrodes and are trapped on the positive electrode surface by electrostatic attraction. In Step 3, when the microbes approach the positive electrode, whose surface is modified by nanowires, they will enter the vicinity of the nanowire tip structure where an enhanced localized electric field exists and become inactivated by electroporation. **c** Schematics of the copper-phosphide-nanowire-modified copper plate ($Cu_3PNW$-Cu) electrode synthesis: copper hydroxide nanowire ($Cu(OH)_2NW$) synthesis on the copper surface by electrochemical oxidation and $Cu_3PNW$ synthesis by phosphidation. **d**, **e** Scanning electron microscope (SEM) and optical images showing the $Cu(OH)_2NW$-modified copper plate ($Cu(OH)_2NW$-Cu) electrode (**d**) and the $Cu_3PNW$-Cu electrode (**e**). Scale bars for inset figures in (**d**) and (**e**) are 3 cm.

powered by the V-TENG. In Step 3, when the microbes approach the surface of the $Cu_3PNW$-modified positive electrode by electrostatic attraction, they will enter the region of the enhanced localized electric field and be effectively inactivated by electroporation.

The $Cu_3PNW$ has been confirmed as a feasible material for nanowire-assisted electroporation disinfection owing to its high conductivity, robust physical structure, and being chemically inert[20]. The $Cu_3PNW$ used in this study was synthesized through a simple and scalable two-step process (Fig. 1c)[20]. The precursors of the $Cu_3PNWs$, copper hydroxide nanowires ($Cu(OH)_2NWs$), were first synthesized on a copper plate (6 cm × 2 cm) using an electrochemical anodization process with a fixed current density (5 mA/cm²) in a NaOH solution (3.0 M) for 30 min[20,32]. After anodization, the color of the copper plate changed from reddish brown to blue (Fig. 1c, d) and the $Cu(OH)_2NWs$ were rooted vertically and uniformly on the electrode surface with lengths of ~5 μm and diameters of ~50 nm (Fig. 1d and Supplementary Fig. 2). Then the prepared $Cu(OH)_2NW$-modified copper plate ($Cu(OH)_2NW$-Cu) electrode was placed in sodium hypophosphite at 120 °C for 90 min with Ar flushing for it to undergo a phosphidation process. After phosphidation, the $Cu(OH)_2NWs$ were then converted to the $Cu_3PNWs$ while retaining a similar morphology (Fig. 1e and Supplementary Fig. 3) but with the color of the electrode changing to black (Fig. 1e).

**Construction and output performance of the V-TENG.** A vertical contact-separation-type TENG was developed as the vibration harvesting system to drive the air disinfection system (Fig. 2a, left and Supplementary Fig. 4). Made from acrylic, the vibration-driven TENG has a three-layer structure with springs used to support and connect each layer. On the top and bottom layers, Al, the positive triboelectric material, was attached to the surface to serve as one electrode of the TENG. On the middle layer, Al was also attached to the surface of both sides before

covering with a perfluoroalkoxy (PFA) film. The PFA film is a negative triboelectric material and the PFA-covered Al electrode serves as the other electrode of the TENG[33]. During operation, the TENG is placed on a shaker with aimed frequency and amplitude to mimic the actual operating conditions of the ventilator in a building. The middle layer of the TENG will vibrate due to the vibration of the ventilator operating with a frequency of ~30 Hz[30,34]. The vibrating middle layer may contact and then separate from the top/bottom layers and AC power will be generated during this contact-separation process due to the coupling effect of triboelectrification and electrostatic induction[35]. The working mechanisms of the vibration-mode TENG was shown in Fig. 2a (right). Due to the different triboelectric polarities of the two triboelectric surfaces, after contacting each other, positive and negative electric charges were created on the Al surface of the top/bottom layers and the PFA surface of the middle layer, respectively. As the middle layer separated from the top/bottom layer, an electrical current was generated due to the redistribution of the free charges between the two electrodes. Thus, when the middle layer vibrates periodically, a contact-separation process occurred between the electrodes on the middle and top/bottom layers, thereby generating an AC output. Considering the small amplitude of the ventilator (normally <1 mm) during operation, the resonance frequency of the TENG middle layer was developed to be the same as the vibration frequency of the ventilator (~30 Hz) using a specific design to harvest the energy of mechanical vibration to provide sufficient electricity for disinfection (see details in Supplementary Note 1). In addition, to guarantee the durability of the TENG, the top and bottom layers were fixed using acrylic to create a closed structure (Fig. 2a and Supplementary Fig. 4).

To investigate its output performance, the V-TENG was fixed on a shaker for frequency sweep experiments (from 1 to 40 Hz) at a constant amplitude (500 μm). Considering the amplitude of the applied vibration (500 μm) less than the gap distance (2 mm) between the two electrodes, contact can only occur if the

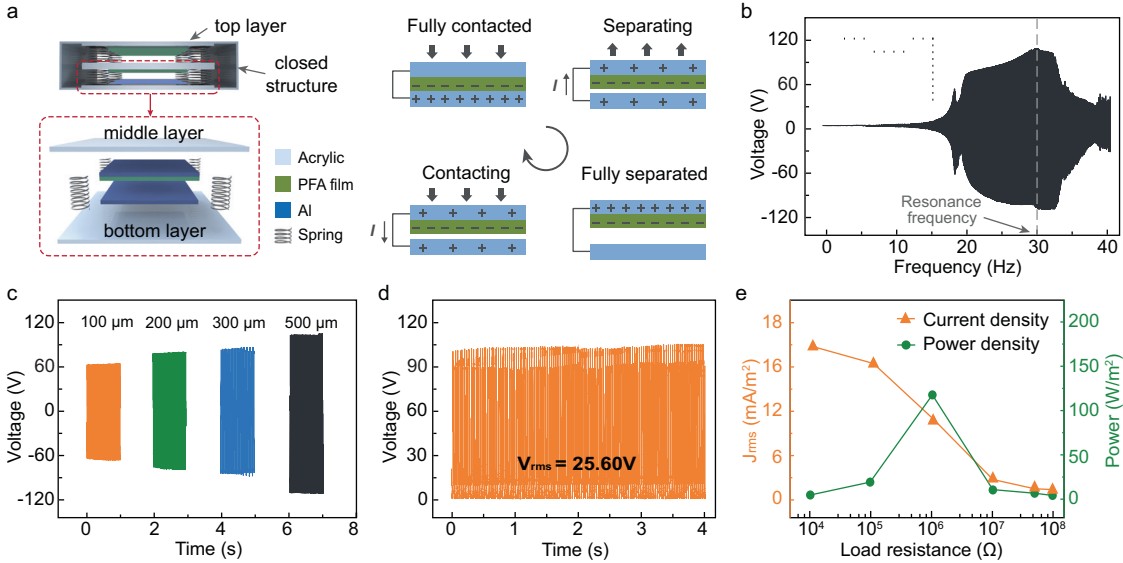

**Fig. 2 Construction and output of the vibration-driven triboelectric nanogenerator (V-TENG). a** Schematics showing the structure (left) and the working mechanism (right) of the V-TENG. **b** Output voltages of the bottom and middle layers of the V-TENG with a vibration frequency sweep from 1 to 40 Hz at a constant amplitude (500 μm). The largest output occurs when the vibration frequency is the same as the resonance frequency of the V-TENG (30 Hz). **c** Output voltages of the V-TENG at various amplitudes (from 100 to 500 μm) and a constant vibration frequency (30 Hz). **d** Output voltages of the V-TENG after rectification at a constant amplitude (500 μm) and vibration frequency (30 Hz). **e** The root mean square current density ($J_{rms}$) and power (*P*) of the V-TENG after rectification with various load resistances (from $10^4$ to $10^8$ Ω) when operated at a constant amplitude (500 μm) and vibration frequency (30 Hz).

amplitude of the middle layer increases due to resonance. As shown in Fig. 2b, the output voltage between the middle and bottom layers was obvious for frequencies ranging from 20 to 33 Hz and achieved its highest output at a vibration frequency of 30 Hz (with a peak-to-peak voltage of 227 $V_{p-p}$). This confirmed that the resonance vibration was capable of generating both large amplitudes and sufficient output for our purposes. The output voltages of the electrodes between the middle and top layers showed a similar tendency and also achieved their highest output at a vibration frequency of 30 Hz (Supplementary Fig. 5). In addition, the output of the V-TENG when driven by various amplitudes (from 100 to 300 μm) was also tested. Based on the frequency sweep experiments, the V-TENGs achieved their highest output at the resonance frequency (30 Hz) for all tested amplitudes (Supplementary Fig. 6). When the vibration frequency was fixed at 30 Hz, the V-TENG can generate an output with a voltage of 130, 162, 182, and 227 $V_{p-p}$ at the amplitude of 100, 200, 300, and 500 μm, respectively (Fig. 2c). After rectification, the output of the V-TENG was tuned to be DC with an open-circuit voltage of 104 V and a short-circuit current of 62 μA (Fig. 2d and Supplementary Fig. 7). This is sufficient for charging the negative electrode, maintaining the electric field between the positive and ground electrodes, and achieving the electroporation disinfection. The output performance of the V-TENG with various load resistances (from $10^4$ to $10^8$ Ω) after rectification was also tested: a maximum of 125 W/m² of power was achieved when with a load of $10^6$ Ω, indicating the feasibility of using our V-TENG to drive an air disinfection system (Fig. 2e).

**Disinfection performance investigation**. The disinfection performance of the V-TENG-powered RV-disinfection method for air-transmitted microbes was evaluated using a prototype to simulate the actual applications (Supplementary Fig. 8). In our work, feed solutions containing a high concentration of bacteria or viruses were added into a super-fine air compressor nebulizer and bacterial or viral bioaerosols were generated by the nebulizer

to flow through the duct. The airflow rate in the duct was controlled to be in the range from 0.5 to 2 m/s using compressed gas. The humidity was fixed at 30% using another nebulizer to generate water aerosols in the duct. Both the airflow rate and the humidity were monitored in real-time using integrated sensors. After flowing through the disinfection filter, the airflow with bacteria or viruses was collected in a narrow mouth bottle containing 500 mL of sterilized deionized (DI) water (Supplementary Figs. 1 and 8). All the bacteria or viruses that remained in the collecting water can be used for future quantification. The V-TENG was operated at a constant amplitude (500 μm) and vibration frequency (30 Hz) to drive the RV-disinfection system. The microbial concentration in the air before and after flowing through the RV-disinfection system was tested using standard spread plating (bacteria) and double agar layer (viruses) methods to analyze the disinfection efficiency (details of the microbial quantification process in Supplementary Fig. 9)[36,37].

The disinfection performance was evaluated using two model bacteria and one model virus. The model bacteria were *Escherichia coli* (*E. coli*) and *Bacillus subtilis* (*B. subtilis*), which represent Gram-negative and Gram-positive species, respectively. The model virus was MS2, an F+ bacteriophage of *E. coli* often used as a process surrogate for human enteric viruses. As shown in Fig. 3a, the V-TENG-powered RV-disinfection system achieved complete disinfection of *E. coli* (>4.1 log removal efficiency corresponding to >99.99% inactivation with no detection of live *E. coli* in the airflow) at airflow rates ranging from 0.5 to 2 m/s. While almost no *E. coli* were inactivated (< 0.21 log removal efficiency) when the output power of the V-TENG was disconnected. Furthermore, the Gram-positive bacteria (*B. subtilis*) was completely inactivated at airflow rates from 0.5 to 1.5 m/s, showing similar to the disinfection performance for *E. coli* (Gram-negative; Fig. 3b). As the airflow rate increased, the disinfection efficiency for *B. subtilis* slightly decreased and showed a lowered disinfection efficiency: 3.9 log removal efficiency (>99.99% inactivation) at an airflow rate of 2 m/s. This lowered disinfection performance is likely due to the

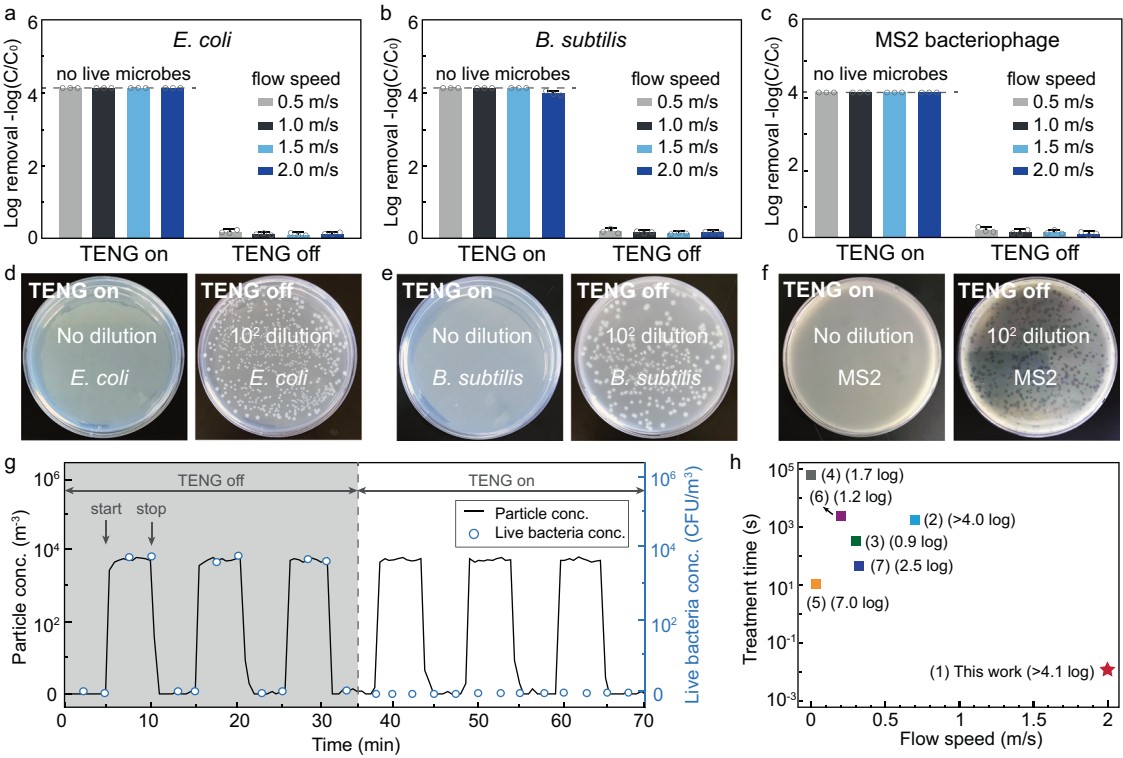

**Fig. 3 Disinfection performance of the V-TENG-powered RV-disinfection method for air-transmitted microbes. a–c** Disinfection efficiency for *Escherichia coli* (*E. coli*; Gram-negative bacteria (**a**)), *Bacillus subtilis* (*B. subtilis*; Gram-positive bacteria (**b**)), and MS2 (virus; **c**) using the RV-disinfection method at various airflow rates (from 0.5 to 2 m/s) powered by V-TENGs. **d–f** Plating results showing the concentration of *E. coli* (**d**), *B. subtilis* (**e**), and MS2 (**f**) after passing through RV-disinfection with or without the power from V-TENGs. **g** The disinfection performance of the RV-disinfection method for treating intermittent bacterial bioaerosols containing *E. coli*. The nebulizer with the feed solution was turned on (start) and then off (stop) for 5 min to generate an intermittent microbial bioaerosol and this periodically turning on and off leads to a periodic change of the microbial concentration. **h** Disinfection-performance comparison of RV-disinfection method using values in the literature describing other air disinfection methods. The plot shows speed of airflow with respect to treatment time for all the air disinfection methods in the comparison: (1) RV-disinfection (this work; complete disinfection, >4.1 log removal), (2) metal-organic-framework-modified filter (ref. [9]; complete disinfection, >4.0 log removal), (3) TiO$_2$-film (ref. [12]; 0.9 log removal), (4) ZnO-Al$_2$O$_3$-filter (ref. [13]; 1.7 log removal), (5) Fe-mesh (ref. [15]; 7.0 log removal), (6) vacuum UV (ref. [10]; 1.2 log removal), and (7) UV-LED (ref. [11]; 2.5 log removal). Details of the conditions for the disinfection experiments (applied materials, disinfection mechanisms, energy demands, and microbial strain) are given in Supplementary Table 1. The final inactivation efficiencies are limited by the initial microbial concentration. In (**a–g**), V-TENGs were operated at a constant amplitude (500 μm) and vibration frequency (30 Hz), and in (**d–g**), the airflow rate was fixed at 2 m/s. Dashed lines indicate that all microbes are inactivated, and no live microbes can be detected. The error bars represent the standard deviation of three replicate measurements.

thicker layer of peptidoglycan on the cell membrane of the Gram-positive bacteria[38]. For the model virus (MS2), the V-TENG-powered RV-disinfection achieved similar disinfection performance to that for *E. coli* (Fig. 3c). All the MS2 were inactivated completely (>4.1 log removal efficiency corresponding to >99.99% inactivation with no detection of live MS2 in the airflow) at airflow rates from 0.5 to 2 m/s. The plating results (Fig. 3d–f) showed a comparison between the highly efficient microbial (*E. coli*, *B. subtilis*, and MS2) inactivation by the RV-disinfection with V-TENG output (right) and control groups without V-TENG output (left) when the airflow rate was set to 2 m/s. In addition, when disinfection was carried out at smaller vibration amplitudes for the V-TENG (from 100 to 400 μm), the power outputs generated from V-TENGs were sufficient to drive the RV-disinfection system to enable high-performance disinfection of bacteria (*E. coli*) and viruses (MS2). At a 2 m/s airflow rate, no live microbes can be detected at amplitudes ranging from 200 to 400 μm, indicating the complete disinfection. Furthermore, >3.7 log microbial removal efficiency (>99.98% microbes were inactivated) was achieved at an amplitude of only 100 μm at the same airflow rate (2 m/s; Supplementary Fig. 10).

Air-transmitted microbes are commonly in an unstable and changeable concentration in the actual situation, thus, the disinfection performance of our RV-disinfection method for treating intermittent microbial bioaerosols in the airflow was evaluated. After fixing the airflow rate and the humidity at 2 m/s and 30%, respectively, the nebulizer with the feed solution of bacteria (*E. coli*) or viruses (MS2) was turned on (start in Fig. 3g) for 5 min and then off (stop in Fig. 3g) for 5 min to generate an intermittent microbial bioaerosol. This periodically turning on and off the nebulizer leads to a periodic change of the microbial concentration. Without the power from the V-TENG, the live bacterial concentration in the airflow was similar to the concentration of the total amount of bacterial cells (i.e., particle concentration), whereas when powered by the V-TENG, no live bacteria can be detected in the intermittent bioaerosols (Fig. 3g). In addition, when treating an intermittent bioaerosol containing viruses, the RV-disinfection method also achieved a similar disinfection performance: no live viruses can be detected in the intermittent bioaerosols after passing through the system (Supplementary Fig. 11). The disinfection performance of the RV-disinfection method was also evaluated by feeding it airflows containing microbes with various concentrations. With the power from V-TENGs, all the *E. coli* and MS2 in different concentrations (from $10^2$ to $10^8$ colony-forming unit/m$^3$ of *E. coli* and from $10^3$ to $10^9$ plaque-forming unit/m$^3$ of MS2) in the bioaerosols

were completely inactivated (Supplementary Figs. 12 and 13). Furthermore, the disinfection performance under various humidity was evaluated. All the tested bacteria (*E. coli*) and viruses (MS2) can be completely inactivated in a wide range of humidity (from 30% to 80%) at a fast airflow rate (2 m/s; Supplementary Fig. 14).

The microbial disinfection performance of RV-disinfection, involving the complete disinfection of *E. coli* and MS2 at an airflow rate of 2 m/s (corresponding to a treatment time of 0.025 s), is the best performance seen so far based on a review of the literature including airflow rates and treatment times (Fig. 3h and Supplementary Table 1). According to the literature review, one commonly used air disinfection method is separating the microbes on a filter followed by applying antibacterial nanomaterial to achieve disinfection (metal-organic-framework-modified filter; TiO$_2$-film; ZnO-Al$_2$O$_3$)[9,12,13]. Although the separation process guarantees a relatively fast airflow rate (up to 0.7 m/s), the following disinfection process is usually time-consuming (>30 min). In the other common type of the air disinfection method, the airflow carrying the microbes will pass through the antibacterial filters (Fe-mesh)[15] or radiations from UV lamps[10,11] without separation to shorten the treatment time (as low as 10 s). However, this approach leads to a decreased airflow rate (ranging from 0.05 to 0.3 m/s). Hence, compared with previously reported air disinfection methods that have a long treatment time and/or low airflow rate, our RV-disinfection enables complete air disinfection at a treatment time of 0.025 s and an airflow rate of 2 m/s. It is clear that our approach is much faster and has great potential for use as an effective air disinfection method.

**Contribution of the macro-mesh negative electrode**. One essential precondition for achieving the highly efficient air disinfection at fast airflow is using a macro-mesh negative electrode to charge the microbes to enable the accelerated microbial trapping on the positive electrode surface. To demonstrate the contribution of the macro-mesh negative electrode in charging the microbes, disinfection performance was compared between the two operation models, one using charging (charge-model) and the other not (no-charge-model; Fig. 4a). Compared with the charge-model using the macro-mesh electrode for charging the microbes (Fig. 4a; up), in the no-charge-model, the previous ground electrode was negatively charged, taking the place of the macro-mesh electrode, so no electric charging was applied to the passing microbes (Fig. 4a; down). After flowing through the charge-model RV-disinfection system with the negatively charged macro-mesh electrode, high disinfection performance was achieved for both bacteria (*E. coli* and *B. subtilis*) and viruses (MS2): > 3.9 log removal efficiency (>99.99% microbial inactivation) at a fast airflow (2 m/s; Fig. 4b). However, with the no-charge-model RV-disinfection, only ~0.5 log removal efficiency was achieved under the same operating conditions. Without the charging process, only trace charge exists on the microbes. Although the enhanced localized electric field exists near the surface of the nanowire-modified positive electrode in the no-charge-model, these microbes cannot approach the positive electrode surface due to the weak electrostatic attraction and short traveling time (<0.025 s). Because electroporation disinfection only occurs when microbes approach the electrode surface, the antibacterial efficiency is poor in the no-charge-model system. The significantly lowered disinfection performance of the no-charge-model RV-disinfection is strong evidence indicating the importance of the macro-mesh negative electrode for charging the microbes during rapid air disinfection.

A simulation of particles (diameters from $10^{-2}$ to 10 μm) flowing through the macro-mesh electrode was carried out to quantify the electrode's contact efficiency (percentage of particles contacting the electrode; see simulation details in Supplementary Table 2). The applied four-layer macro-mesh electrode achieved >99.1% and >99.6% contact efficiency of particles with diameters from 0.02 to 0.1 μm (representing viruses) and diameters from 0.5 to 4 μm (representing bacteria), respectively (Fig. 4c). Decreasing the number of the layers significantly lowered the contact efficiency, especially for the particles with smaller diameters (i.e., viruses; Fig. 4c). The disinfection performance using different layers of electrodes (from 1 to 4 layers) also confirmed that lowered contact efficiency led to poorer disinfection performance (Fig. 4d). In addition, the macro-mesh structure was confirmed to be the ideal electrode structure owing to it giving the highest contact efficiency compared to other structures (column and slope; Supplementary Fig. 15).

After flowing through the electrode, the charges carried by each single microbe were measured and calculated. Compared with the trace charges carried by microbes in the air before passing through the electrode ($<10^{-15}$ C), after contact with the negatively charged electrode, one single *E. coli* cell or MS2 particle carried charges of $6.1 \times 10^{-10}$ or $7.2 \times 10^{-12}$ C, respectively (Fig. 4e)[39]. In addition, the pressure drops due to the RV-disinfection systems with the macro-mesh electrode were measured and compared to that caused by a HEPA filter, the most commonly used air-transmitted pathogen removal method. When using the macro-mesh electrode, the pressure drop was low but increased slightly as the airflow rate increased: pressure drops of 2, 6, 13, and 24 Pa were seen at airflow rates of 0.5, 1, 1.5, and 2 m/s, respectively (Fig. 4f). When using a HEPA filter, even at 0.5 m/s of airflow, a 105 Pa of pressure drop was detected, which is >50 times higher than that for the macro-mesh electrode under the same operating conditions. When the airflow rate was increased to 1 m/s, a 200 Pa of pressure drop was detected, reaching the detection limit of the sensor (Fig. 4f). The simulated airflow field before and after flowing through the macro-mesh electrode also indicated only a limited impact of the electrode on the airflow (Fig. 4g and see simulation details in Supplementary Table 2).

In the RV-disinfection process, when airflow passes through the negatively charged macro-mesh electrode, the bacteria and viruses in the air will first come into contact with the electrode surface and become charged. These negatively charged microbes will then flow between the integrated positive/ground electrodes and be immediately trapped on the positive electrode surface by the electrostatic attraction to achieve disinfection even at a fast airflow rate (2 m/s; see calculation details in Supplementary Note 3). This accelerated charging/trapping process overcomes the speed limitations of the nanowire-assisted electroporation, where the time-consuming process for microbes to approach the electrode surface slows down disinfection, and thus, our approach enables the rapid air disinfection (~m/s; >200 folds the flow rate seen in water disinfection).

**Disinfection analysis of the nanowire-modified positive electrode**. To investigate the importance of the nanowire structure for enhancing the localized electric field, disinfection performance between positive electrodes made of Cu$_3$PNWs and copper phosphide nanoparticles (Cu$_3$PNPs) was evaluated and compared using bacteria (*E. coli* and *B. subtilis*) and viruses (MS2). The Cu$_3$PNPs-modified copper plate (Cu$_3$PNP-Cu) electrode was fabricated by exposing the Cu(OH)$_2$NW-Cu electrode to a higher temperature of 180 °C for 2 h during the phosphidation process. Instead of maintaining the nanowire structures in the same way as at the lower temperature (120 °C, Fig. 5a; left), the Cu(OH)$_2$NWs melted at 180 °C, as such, nanoparticles formed

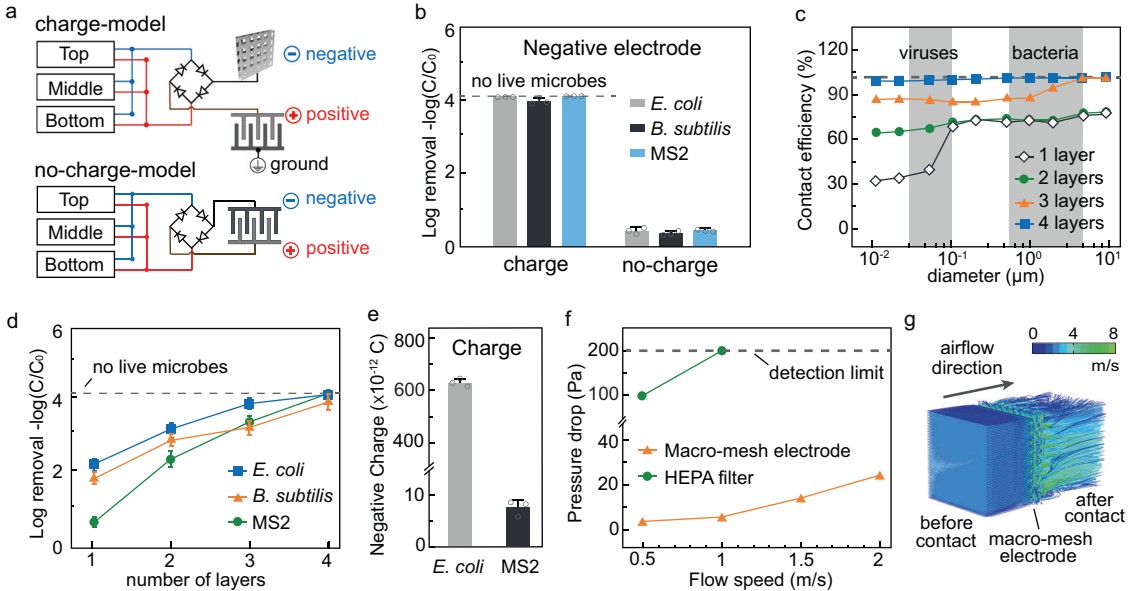

**Fig. 4 Contribution of the macro-mesh negative electrode. a** Schematics of the circuits for the charge-model (with the macro-mesh electrode; top) and no-charge-model (without the macro-mesh electrode; bottom) to investigate the contribution of the macro-mesh electrode in the RV-disinfection system. **b** Disinfection performance of the charge-model and the no-charge-model RV-disinfection systems for *E. coli*, *B. subtilis*, and MS2. Dashed lines indicate that all microbes are inactivated, and no live microbes can be detected. **c** Simulation of the contact efficiency (percentages of particles with various diameters from $10^{-2}$ to $10\,\mu m$ that come into contact with the electrode) after flowing through the mesh electrode with various numbers of layers (from 1 to 4 layers). Dashed lines indicate 100% contact efficiency. **d** Disinfection performance of the RV-disinfection system using a macro-mesh negative electrode with various numbers of layers (from 1 to 4 layers). Dashed lines indicate that no live microbes can be detected. **e** Quantitative measurement of the charges carried by each *E. coli* and MS2 after flowing through the macro-mesh negative electrode. In (**b–e**), the airflow rate was fixed at 2 m/s, and the V-TENG was operated at a constant amplitude (500 μm) and vibration frequency (30 Hz). **f** Pressure drops in airflows after passing through the RV-disinfection system with the macro-mesh electrode or with the high-efficiency particulate air (HEPA) filter at various airflow rates (from 0.5 to 2 m/s). Dashed lines indicate the detection limit (200 Pa). **g** Simulation of the airflow field before and after flowing through the RV-disinfection system with the macro-mesh electrode at an airflow rate of 2 m/s. The error bars represent the standard deviation of three replicate measurements.

uniformly on the surface of the copper electrodes (Fig. 5a; right). Formations of $Cu_3P$ for both NWs and NPs samples were confirmed from X-ray diffraction (XRD) and X-ray photoelectron spectroscopy (XPS) analysis (Supplementary Fig. 16). When using the $Cu_3PNWs$-Cu electrode, the RV-disinfection system achieved >3.9 log removal efficiency (>99.99% inactivation) for both bacteria (*E. coli* and *B. subtilis*) and viruses (MS2) at 2 m/s of airflow. However, when using the $Cu_3PNP$-Cu electrode, the RV-disinfection system was ineffective on the tested model microbes (<0.15 log removal efficiency) under the same operating conditions. This great difference in disinfection performance between the $Cu_3PNW$-Cu and $Cu_3PNPs$-Cu electrodes is caused by the unique geometry of nanowires that enhance the localized electric field to achieve electroporation disinfection. Furthermore, the electric field simulation in Fig. 5c showed that when powered by a V-TENG after rectification (100 V DC), the electric field near the $Cu_3PNW$ tip was enhanced (>$10^8$ V/m), which is sufficient for microbial electroporation disinfection (see simulation details in Supplementary Table 3). When applied with similar operating conditions, the electric field near $Cu_3PNPs$ is insufficient for disinfection according to the simulation results (Supplementary Fig. 17).

To further demonstrate the disinfection mechanisms at work during RV-disinfection operation, other potential disinfection mechanisms including chemical oxidation, intracellular reactive oxygen species (ROS) generation, and toxicity of the released copper ion ($Cu^{2+}$) from the positive electrode were also investigated. Firstly, chemical oxidation was evaluated using radical scavengers that can react quickly with generated oxidative species (normally ·OH and ·$O_2^-$) to eliminate oxidation[40].

Isopropanol (IPA; ·OH scavenger; ~1 mM) and benzoquinone (BQ; ·$O_2^-$ scavenger; ~1 mM) were added to the microbial feed solution, respectively, and included in the bioaerosol generated by the nebulizer. After flowing through the RV-disinfection system at 2 m/s of airflow, the added radical scavengers showed no impact on the disinfection efficiency for both bacteria (*E. coli* and *B. subtilis*) and viruses (MS2), indicating no chemical oxidation during the RV-disinfection (Fig. 5d). Secondly, when bacteria come into contact with the electrode surface, intracellular ROS may be generated due to charge transfer between the bacteria and electrode, which may cause inactivation[19,41]. Compared with a positive control (0.1 mM $H_2O_2$), only trace intracellular ROS was generated in bacteria (<3% intensity of the positive control), indicating the minor contribution of ROS to RV-disinfection (Fig. 5e). Thirdly, the airflow (0.5 m³) containing $Cu^{2+}$ released from the positive electrode after flowing through the RV-disinfection system was collected in 500 mL of DI water, and the concentration of $Cu^{2+}$ in the collecting water was evaluated. Owing to the robust physical structure and chemically inert features of the $Cu_3PNW$-Cu electrode, only 2 μg/L of $Cu^{2+}$ was detected in the collecting water (~3 orders of magnitude lower than the standard for safe drinking water; 1000 μg/L) This indicated the ineffectiveness of the released $Cu^{2+}$ on RV-disinfection (Fig. 5f)[20]. In addition, the released $Cu_3P$ was confirmed to be ineffective for bacterial inactivation due to its low concentration based on the biocompatibility test (Supplementary Fig. 18).

The contribution of Joule heating to disinfection was also evaluated by measuring the surface temperature of the nanowire-modified positive electrodes (Supplementary Fig. 19). No

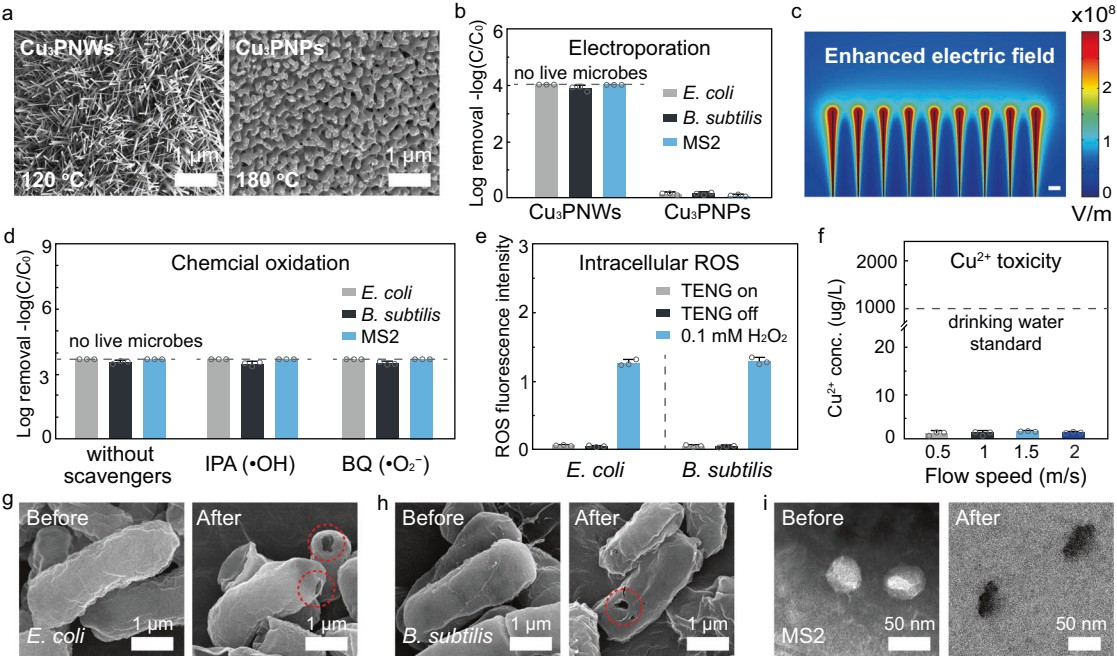

**Fig. 5 Disinfection mechanism investigation into the nanowire-modified positive electrode. a** SEM images showing the Cu(OH)$_2$NW-Cu electrode after the phosphidation process at 120 and 180 °C forming the Cu$_3$P with the geometry of nanowire (left) and nanoparticle (right), respectively. **b** Disinfection performance of the RV-disinfection system using nanowire- and nanoparticle-modified electrodes, showing the enhanced performance of the nanowire structure. Dashed lines indicate that no live microbes can be detected. **c** Simulation of electric field distribution near the surface of Cu$_3$PNW (diameter, 50 nm; length, 5 μm) driven by the V-TENG, showing the enhancement of the localized electric field. The scale bar is 500 nm. **d–f** Investigation of other mechanisms that may contribute to RV-disinfection. **d** Disinfection performance of RV-disinfection system with or without the added scavengers (isopropanol; IPA and benzoquinone; BQ). Dashed lines indicate that no live microbes can be detected. The radical scavengers can react quickly with specific oxidizing agents (IPA for ·OH and BQ for ·O$_2^-$) to eliminate the chemical oxidation. **e** Measurement of the intracellular reactive oxygen species (ROS) caused by the physiological change transfer. In (**d**) and (**e**), the airflow rate was fixed at 2 m/s. **f** Concentrations of copper ions (Cu$^{2+}$) after 0.5 m$^3$ of airflow collected in 500 mL of water, showing no contribution of Cu$^{2+}$ to RV-disinfection. Dashed lines indicate the Cu$^{2+}$ concentration standard for safe drinking water (1000 μg/L). **g–i** SEM images of bacteria and transmission electron microscope (TEM) images of viruses before and after flowing through the RV-disinfection system, showing the disinfection mechanism of electroporation. SEM images of *E. coli* (Gram-negative bacteria (**g**)) and *B. subtilis* (Gram-positive bacteria (**h**)) before (left) and after (right) RV-disinfection operation. After operation, electroporated pores were found generated on the bacterial membranes. **i** TEM images of viruses (MS2) before (left) and after (right) RV-disinfection operation. After operation, the MS2 capsid was damaged and the inside was stained. In (**g–i**), the airflow rate was fixed at 2 m/s. The V-TENG was operated at a constant amplitude (500 μm) and vibration frequency (30 Hz). The error bars represent the standard deviation of three replicate measurements.

significant temperature fluctuation occurred during operation, indicating no significant contribution of Joule heating to microbial inactivation. In addition, considering the relatively long contact time (from several seconds to minutes) needed for thermal disinfection, the Joule heating is not likely to contribute to microbial inactivation due to the short contact time (0.025 s)[15,42]. When microbes were attached to the nanowire-modified electrodes, the sharp structure of the nanowire may cause mechanical damage to the microbes and contribute to disinfection. However, to achieve microbial inactivation based on mechanical damage, a relatively long contact time from several seconds to minutes is commonly necessary[43,44]. Due to the short contact time in our work (<0.025 s), the mechanical damage may only have little contribution to microbial inactivation.

The disinfection mechanism was further confirmed to be electroporation based on a microbial morphology analysis using scanning electron microscopy (SEM; for bacteria) and transmission electron microscopy (TEM; for viruses). After RV-disinfection at a fast airflow (2 m/s), both Gram-negative (*E. coli*; Fig. 5g) and Gram-positive (*B. subtilis*; Fig. 5h) bacteria showed electroporation pores (~100 nm) on the surface[17]. Commonly, after disinfection treatment based on the mechanisms of chemical oxidation, charge transfer, ion toxicity, Joule heating, and/or mechanical damage, the bacteria will collapse due

to the damaged outer structure[41,43–46]. Nevertheless, after RV-disinfection, the cell structure still maintained only with specific electroporation pores (~100 nm) on the bacterial surface. This could be additional strong evidence confirming the disinfection mechanism to be electroporation. TEM was used to characterize MS2 before and after RV-disinfection due to the smaller viral size (~50 nm). A negative staining method was used to investigate whether the capsid integrity of MS2 was maintained[47]. After RV-disinfection, the MS2 showed dark contrast, indicating that the inside of the MS2 was stained and the capsids were damaged (Fig. 5i). In summary, based on the above disinfection mechanism investigation, the cause responsible for microbial inactivation in our RV-disinfection system is confirmed to be electroporation.

## Discussion

In summary, we developed a high-performance self-powered RV-disinfection method for the rapid disinfection of air-transmitted bacteria and viruses using energy harvesting devices (V-TENGs) to supply sufficient electricity from the weak mechanical vibrations (amplitude as low as 100 μm). A nanowire-assisted electroporation mechanism was applied for air disinfection and was assisted by the accelerated charging and trapping of microbes to overcome this method's previous speed limitations. High

disinfection performance was achieved: >99.99% bacteria and viruses were inactivated at a fast airflow rate (2 m/s), corresponding to a treatment time of 0.025 s, while maintaining only a low pressure drop (24 Pa). The promising performance of RV-disinfection on bacteria and viruses gives this approach great potential to fill the vital need for a rapid indoor air disinfection system to protect people against the air-transmitted microbial infection with no need for external energy input.

## Methods

**Electrode fabrication and characterization**. The macro-mesh negative electrode was made of stainless-steel foil (Alfa Aesar, 0.5 mm thick) and was cut into 6 cm × 6 cm to create a square electrode with macro-pores (5 mm × 5 mm) distributed evenly throughout the electrode to give a macro-mesh structure. Varying numbers of layers (from 1 to 4) of the prepared macro-mesh electrodes were fixed with a distance of 5 mm between them in an acrylic holder. The $Cu_3PNW$-Cu positive electrode was prepared using a two-step process. Firstly, the Cu foil (Alfa Aesar, 0.5 mm thick) was cut into a rectangular shape with a size of 6 cm × 2 cm and washed first with 1 M HCl (Sigma) and then subsequently with DI water to remove any surface impurities. The cleaned Cu foil was then anodized in 3.0 M NaOH solution (Sigma) for 30 min under 5 mA/cm$^2$ to fabricate the $Cu(OH)_2$NWs on the surface. The current was provided using a DC power supply (UNI-T, UTP1303). Secondly, to prepare the $Cu_3PNWs$, excess sodium hypophosphite (Sigma) was placed at the center of a tube furnace. After flushing with Ar for 15 min, the center of the furnace was elevated to 300 °C and the electrode with $Cu(OH)_2$NWs was placed downstream of the furnace where the temperature was 120 °C. After 90 min, the furnace cooled down naturally to room temperature under an Ar atmosphere and the $Cu_3PNW$-Cu electrode was obtained. The ground electrode was made of stainless-steel foil (Alfa Aesar, 0.5 mm thick) and was a similar size to the positive electrode (rectangular electrode with a size of 6 cm × 2 cm). Three positive electrodes and three ground electrodes were integrated into a parallel structure at a distance of 1 cm from each other and were fixed in an acrylic holder. The morphology of the $Cu_3PNW$-Cu electrode was then characterized using an SEM (JEOL, JSM-7500F) with an accelerating voltage of 15 kV. The material components were analyzed by XRD (PANalytical, Alpha 1 MPD). The chemical compositions were analyzed by an XPS using an Axis Ultra instrument (Kratos Analytical, K-alpha) with ultrahigh vacuum (<10$^{-8}$ Torr) and a monochromatic Al Kα X-ray source.

**V-TENG fabrication, operation, and characterization**. The V-TENG used a three-layer structure (top, middle, and bottom layers) and was made of acrylic. The Al foil (Alfa Aesar, 25 μm thick) was cut into 4 cm × 4 cm squares and fixed to the surface of the top and bottom layers. For the middle layer, Al was also attached to the layer surface, but this was then covered by a PFA film (DuPont, 80 μm thick). Al and PFA on the middle layer were the same size (4 cm × 4 cm) as the Al on the top and bottom layers. The total mass of the middle layer was carefully set to 20 g. Four springs with a spring constant of 82.712 N/m each provided a 2 mm gap between the top/middle and the bottom/middle layers, respectively. The bottom and the top layers were fixed using acrylic to create a closed structure. During operation, the V-TENG was fixed to a vibration input source (The Labworks, ET-126 Electrodynamic Transducer) with a sweeping (from 1 to 40 Hz) or a fixed (30 Hz) frequency and various amplitudes (from 100 to 500 μm). The outputs (e.g., voltage and current) of the V-TENG from contact between the top/middle and the bottom/middle layers before and after rectification were measured using an oscilloscope (Tektronix, DOP 3052).

**RV-disinfection system construction**. The macro-mesh negative and positive/ground integrated electrodes were fixed in an acrylic duct (6 cm × 6 cm cross-section) and powered by the output of the V-TENG after rectification. The length of the duct was designed to be 1.4 m to ensure laminar airflow inside (see design details in Supplementary Note 2). A super-fine air compressor nebulizer (Philips) containing feed solutions with a high concentration of bacteria or viruses was used to generate bioaerosol to feed the duct. Airflow rates in the duct were set in the range from 0.5 to 2 m/s using compressed gas while the humidity was fixed at 30% using another nebulizer (Philips) generating water mist flow into the duct. The airflow rate, humidity, particle concentration, and pressure drop were monitored in real-time using the integrated sensors.

**Disinfection performance analysis**. Bacteria (*E. coli*; ATCC 15597 and *B. subtilis*; ATCC 23857) were cultured in Tryptic Soy Broth (TSB, Sigma) to log phase (12 h) and harvested by centrifugation at 1500*g* (HITACHI, RX2 series) before being suspended in DI water. After washing with DI water 3 times, bacterial cells were finally suspended in DI water to serve as the feed solution with a concentration of ~10$^9$ colony-forming unit/mL. Viruses, bacteriophage MS2 (ATCC 15597-B1), were grown with the *E. coli* host on a shaker table set to 50 rpm at 37 °C for 24 h. MS2 was isolated and concentrated using the polyethylene glycol (PEG) precipitation method. A solution of ~10$^{10}$ plaque-forming unit/mL was prepared in DI water to serve as the feed solution. After flowing through the RV-disinfection

system, the airflow (0.5 m$^3$) with bacteria or viruses was collected in a narrow mouth bottle containing 500 mL sterilized DI water. The microbes collected in the water were used for future quantification and their concentrations were tested using standard spread plating (bacteria) and double agar layer (viruses) methods[36,48]. The disinfection efficiency was analyzed according to the following equation (Eq. 2):

$$\text{Efficiency} = -\log_{10}(C/C_0) \qquad (2)$$

where $C$ and $C_0$ are the concentrations of the collected microbes with and without the RV-disinfection. The log removal efficiency can easily express the level of inactivated microbes with high concentration and quantify the disinfection efficiency of those more than 90%[49]. Each sample was serially diluted, and each dilution was plated in triplicate. Bacterial and viral samples were incubated at 37 °C for 12 and 4 h, respectively (see details of microbial quantification in Supplementary Fig. 9).

**Airflow simulation and contact efficiency calculation**. The speed field of the airflow was simulated using the finite element method with Ansys Fluent software. The contact efficiencies (percentages of particles that came in contact with the electrode) after flowing through the negative electrodes with various diameters (10$^{-2}$ to 10 μm) were calculated according to the simulation results. The contact efficiencies of particles flowing through various numbers of layers (from 1 to 4 layers) and various structures of the electrode (macro-mesh, slope, and column) were also simulated. See details of the values used for the simulation in Supplementary Table 2.

**Electric field simulation**. The electric field distribution was simulated by the finite element method using COMSOL Multiphysics. A 3D model of the electrode was set up and a 9 × 9 nanowires array was built with a length of 5 μm and a diameter of 50 nm to demonstrate the electric field around the tip area. The values used in the simulation reflect the real configuration and operating conditions used for the RV-disinfection device testing (see simulation details in Supplementary Table 3).

**Calculation of charges carried by microbes**. Instead of using the positive electrode, only the macro-mesh negative electrode was used for calculating the charges carried by microbes. The nebulizer with feed solutions containing *E. coli* or MS2 generated aerosols that were fed to the duct at a controlled airflow rate (2 m/s) and humidity (30%). Powered by the V-TENG, the airflow with microbes passed through the negative electrode for 2 min, and the charges that remained on the negative electrode ($C_1$) were measured using an electrometer (Keithley, 6514). The microbes in the airflow were collected in 500 mL DI water and their concentrations ($C_m$) were analyzed. For the control sample, only DI water was added to the nebulizer. The aerosol generated by the nebulizer without microbes also passed through the V-TENG-powered negative electrode under the same operating conditions. After the airflow without microbes passed through the negative electrode for 2 min, the charges that remained on the negative electrode ($C_0$) were also measured. The charges carried by microbes ($C$) after passing through the negative electrode were calculated according the following equation (Eq. 3):

$$C = \alpha \times \frac{C_0 - C_1}{500 \times C_m} \qquad (3)$$

where α is the contact efficiency based on the simulation (99.6% and 99.1% for bacteria and viruses, respectively), $C_0$ and $C_1$ are the charges remaining on the negative electrode for airflow without and with microbes passing through, respectively, and $C_m$ is the bacterial or viral concentration in the collecting water.

**$Cu^{2+}$ concentration measurement**. After passing through the RV-disinfection system, airflow (0.5 m$^3$) was collected in 500 mL sterilized DI water. An 1 mL aliquot of the collecting water was dosed in 1 mL HNO$_3$ (2 M; Sigma) and analyzed by the inductively coupled plasma mass spectrometry (ICP-MS, Thermo Scientific, XSERIES 2) to test the $Cu^{2+}$ concentration.

**Intracellular ROS measurement**. The intracellular ROS levels were measured using a fluorescent probe, 2′,7′-dichlorodihydrofluorescein diacetate (DCFH-DA, Beyotime, China)[41]. The bacteria in the airflow passing through the RV-disinfection system were collected in the DI water. A 0.1 mL aliquot of DCFH-DA was dosed in 1 mL of the collecting water under dark conditions for 15 min. The fluorescent intensity was measured by a microplate photometer (Thermo Scientific, Multiskan FC) with 488 and 520 nm as the excitation and emission wavelengths, respectively. For the ROS-positive control group, 0.1 mM of H$_2$O$_2$ treated the bacteria sample (0.5 mL) for 60 min, then the intracellular ROS level in the H$_2$O$_2$ treated bacteria was measured.

**Bacterial and viral sample preparation for SEM and TEM**. Bacterial samples were harvested by centrifugation at 1500*g* for 5 min at 15 °C (HITACHI, RX2 series), and supernatants were removed. Then the bacteria were fixed overnight in the fixative containing 0.1 M phosphate-buffered solution (pH 7.3; Sigma) and 2% glutaraldehyde (Sigma) at 4 °C. Samples were then dehydrated with increasing concentrations of ethanol solutions (50%, 70%, 90%, and 100%; Sigma)

before drying in 100% t-BuOH (Sigma) using a freeze-drying process (ilShin BioBase. TFD 8501). All the bacterial samples were dispersed on a metal grid in preparation for SEM characterization (JEOL, JSM-7500F). A total of 20 µL of the viral samples were pipetted on a TEM grid, then, after a 15 min air-drying process, samples were stained with 1% phosphotungstic acid solution (Sigma) for 1 min. The TEM grids were air-dried for TEM characterization (JEOL, JEM-2100F).

## Data availability
The data that support the findings of this study are available from the corresponding author upon reasonable request.

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

## Acknowledgements

This work was financially supported by Nano Material Technology Development Program (2020M3H4A1A03084600) through the National Research Foundation of Korea (NRF) funded by Ministry of Science and ICT and the GRRC program of Gyeonggi province (GRRC Sungkyunkwan 2017-B05). Z.-Y.H. acknowledges the support from the Korea Research Fellowship Program through the National Research Foundation of Korea (No. 2019H1D3A1A01102903). Z.-Y.H. also thanks the technical support from Dr. Rong Cheng from Renmin University of China and Dr. Jinling Xue from Helmholtz Zentrum München.

## Author contributions

Z.-Y.H., Y.-J.K., and S.-W.K. developed the concept. Z.-Y.H. synthesized the samples and conducted the disinfection measurement and material characterizations. Y.-J.K. fabricated the RV-disinfection system, performed the TENG measurements, and carried out the airflow simulation. D.-M.L. did the electric field simulation. I.-Y.S. and J.H.L. helped with the measurement of pressure drop and particle concentration. Y.D. helped with the HPLC measurement. S.W. helped with the charge calculation. Z.-Y.H. and S.-W.K. analyzed the data and co-wrote the paper. H.-J.Y. and Y.-J.K. provided important experimental insights. All the authors discussed the whole paper.

## Competing interests

The authors declare no competing interests.
