## [Peer Review File. · Nature Communications]

REVIEWER COMMENTS

Reviewer #1 (Remarks to the Author):

Authors developed the new method of inactivating model microbes (gram +/- and MS2 bacteriophage) in the air through the triboelectric-potential with nanowire-grown electrodes. It was hypothesized that the localized electric field near the nanowires and charges induced by the negative electrode play key roles in the rapid inactivation of bacteria and viruses. The manuscript is very well-written and easy to follow. Due to the timeliness of the current pandemic and the potential application of this manuscript, this manuscript will be very interesting for readers in Nature Comm. Therefore, this reviewer support the publication of the manuscript after the revision to address the followings:

1. The authors claim that the inactivation is induced by the electroporation with the localized electric field near the nanowires. It is difficult to conclude with the reported results that the inactivation of microbes is mainly from the localized field. The authors excluded the effects of oxidation and toxicity; however, there are other mechanisms that can inactivate the microbes during contact with nanowires such as Joule heating and mechanical damage. Due to the electric current, the temperature at the edges of nanowires can be significantly increased very fast.¹ Also, the negatively charged microbes move toward the positively charged nanowires having the sharp tips by the electrophoretic force. Under the force, the microbes can be mechanically damaged while they are trapped on top of the nanowires.² Therefore, the authors need to address other possibilities and revise the manuscript.

Reference

- 1) Jin, C. Y.; Li, Z.; Williams, R. S.; Lee, K. C.; Park, I., Localized Temperature and Chemical Reaction Control in Nanoscale Space by Nanowire Array. *Nano Letters* 2011, 11 (11), 4818-4825.
- 2) Valiei, A.; Lin, N.; Bryche, J.-F.; McKay, G.; Canva, M.; Charette, P. G.; Nguyen, D.; Moraes, C.; Tufenkji, N., Hydrophilic Mechano-Bactericidal Nanopillars Require External Forces to Rapidly Kill Bacteria. *Nano Letters* 2020, 20(8), 5720-5727.

2. Figure S8 shows the flow controller at the inlet and the flow sensor behind the electrodes. If the written flow rates in this manuscript are set values of the flow controller at the inlet, the actual flow near the electrodes should be much slower because the cross-section of the inlet looks much smaller than that of the duct. The actual flow rates near the positive electrode in the duct should be used to discuss the speed of inactivation.

3. Simply mentioning COVID-19 and building ventilation systems would be fine as potential applications in the discussion part, but not in the abstract or first paragraph of the introduction as it can mislead readers. There is not enough connection between this study and the specific examples.

4. $-\log(C/C_0)$ is used as y-axes to show the efficacy of the system as the values are drastically different as shown in Figures 3,4 and 5. However, it disturbs the quantitative analysis; for example, <38% inactivation is still a high value, but <0.21 log seems very low (row 220). Please consider changing the y-axes as inactivation % as needed.

5. Reference needs to be added on row 135.

Reviewer #2 (Remarks to the Author):

In this work, authors have reported a rapid disinfection method for inactivating air-transmitted bacteria and viruses using the nanowire-enhanced localized electric field to damage the outer structures of microbes using the triboelectric nanogenerator that converts mechanical vibration to electricity for the air disinfection purpose. These are some questions that the authors need to clarify.

1. Authors claimed that they have reported first self-powered disinfection system. However, this type of system is already reported: *Nano Energy* 2017, 36, 241-249 (A self-powered sterilization system with both instant and sustainable anti-bacterial ability) and *Nano Lett.* 2014, 14, 10, 5603–5608 (Static Electricity Powered Copper Oxide Nanowire Microbicidal Electroporation for Water Disinfection).
2. Formation of Cu₃P nanowires and nanoparticles should be confirmed from XRD and XPS analysis.
3. In the entire manuscript it is not clear which experiments are conducted in nebulizer and which one in ventilator? Were they conducted in same setup or with different apparatus? Authors should clarify those issues.
4. In the Figure 3g, why the live bacterial concentration changes periodically or more specifically decrease when the TENG is off? What is the reason of reducing the bacterial concentration even if electric field is not applied?
5. If the distance between the positive and negative electrode is kept constant in no charge model (similar to the distance between positive and ground electrode) and one of the electrodes is surface modified with nanostructure, it is obvious that local electric field enhancement will also be there. Then why the antibacterial efficiency is poor in case of no charge model? Author should explain this clearly.
6. Authors should present the simulation of electric field distribution for Cu₃P NPs also in order to confirm the level of field enhancement due to these nanostructures.
7. Author should check the TENG assisted antibacterial performance in different range of humidity (30% to 80%) as in real environment humidity varies a lot.
8. In page 5, line 92-93, the authors mentioned that "The RV air disinfection method designed to be deployed in the ducts of indoor buildings to achieve air disinfection during the normal ventilation process." Therefore, the authors should demonstrate the real time application of RV disinfection process embedding the device in the duct of domestic environment.
9. The mechanism of bacterial destruction is not clear in the manuscript. Authors confirmed that chemical oxidation and intercellular ROS generation were not taking place. Then the authors should specifically mention the cause/mechanism responsible for killing bacteria in RV disinfection system.
10. Since the filtered air will go to the environment, authors should also provide the biocompatibility tests of their materials as Cu₃P toxicity is a well-known phenomenon.

Our Responses to the Comments from Reviewers

Triboelectrification induced self-powered microbial disinfection using nanowire-enhanced localized electric field (NCOMMS-20-46987)

Authors would like to thank all reviewers (including referees and editorial board) for the critical reading and valuable comments that have helped us to make the manuscript even more improved.

Reviewer # 1

Authors developed the new method of inactivating model microbes (gram +/- and MS2 bacteriophage) in the air through the triboelectric-potential with nanowire-grown electrodes. It was hypothesized that the localized electric field near the nanowires and charges induced by the negative electrode play key roles in the rapid inactivation of bacteria and viruses. The manuscript is very well-written and easy to follow. Due to the timeliness of the current pandemic and the potential application of this manuscript, this manuscript will be very interesting for readers in Nature Comm. Therefore, this reviewer supports the publication of the manuscript after the revision to address the followings:

Comment 1:

The authors claim that the inactivation is induced by the electroporation with the localized electric field near the nanowires. It is difficult to conclude with the reported results that the inactivation of microbes is mainly from the localized field. The authors excluded the effects of oxidation and toxicity; however, there are other mechanisms that can inactivate the microbes during contact with nanowires such as Joule heating and mechanical damage. Due to the electric current, the temperature at the edges of nanowires can be significantly increased very fast¹. Also, the negatively charged microbes move toward the positively charged nanowires having the sharp tips by the electrophoretic force. Under the force, the microbes can be mechanically damaged while they are trapped on top of the nanowires². Therefore, the authors need to address other possibilities and revise the manuscript.

[1] Jin, C. Y.; Li, Z.; Williams, R. S.; Lee, K. C.; Park, I., Localized Temperature and Chemical Reaction Control in Nanoscale Space by Nanowire Array. *Nano Letters* 2011, 11 (11), 4818-4825.

[2] Valiei, A.; Lin, N.; Bryche, J.-F.; McKay, G.; Canva, M.; Charette, P. G.; Nguyen, D.; Moraes, C.; Tufenkji, N., Hydrophilic Mechano-Bactericidal Nanopillars Require External Forces to Rapidly Kill Bacteria. *Nano Letters* 2020, 20(8), 5720-5727.

Response 1:

We truly appreciate the reviewer's valuable comments, and we understand the reviewer's concern about the potential microbial inactivation mechanisms (i.e., Joule heating and

mechanical damage) during the disinfection process. We have added additional experimental results, reference, and discussion in the revised manuscript to make our statement clear:

Joule heating is the process by which the passage of an electric current through a conductor produces heat. The power of heating generated by an electrical conductor is proportional to the product of its resistance and the square of the current. Thus, the electric current passing through a closed and complete circuit is the precondition to generate Joule heating. However, in the experimental setup of our work, the positive and negative electrodes are not connected with each other, and only trace current signal can be detected during the operation process (several nA). Due to the open-circuit structure of our device, the generated charge will remain on the surface of electrodes (similar to a capacitor) or transfer to the trapped microbes rather than pass through the circuit to generate Joule heating. It is worth noting that the current output in Fig 2 is the short-circuit current of the nanogenerator without any load, which was measured to evaluate the output performance. Such current at the short-circuit operating condition was not the actual current during the disinfection. Furthermore, we measured the bulk temperature of the nanowire-modified positive electrode with or without the power from TENG and found no significant temperature fluctuation occurred during the operation. We have added this experiment and discussion in the revised manuscript and Supplementary Fig. 19 as follows:

<Line 383>

“The contribution of Joule heating to disinfection was also evaluated by measuring the surface temperature of the nanowire-modified positive electrodes (Supplementary Fig. 19). No significant temperature fluctuation occurred during operation, indicating no significant contribution of Joule heating to microbial inactivation.”

<Supplementary Information>

Supplementary Fig. 19 Bulk surface temperatures of the Cu₃PNWs-modified positive electrode with or without the power from TENG for 30 min. The humidity was fixed at 30% with airflow rates ranging from 0.5 to 2.0 m/s. No microbes were applied in this experiment. No significant temperature fluctuation of the electrode occurred during the air disinfection, confirming little contribution of Joule heating to microbial inactivation.

To achieve microbial inactivation using thermal treatment (including Joule heating), normally at least several seconds to minutes of contact time is needed^{1, 2}. In our work, due to the fast air rate, the microbes will pass through the electrodes within 0.025 s, and the contact time between microbes and electrodes will be even shorter. With such short contact time, the Joule heating isn't likely to contribute to microbial inactivation. Furthermore, after the thermal treatment, the morphology of bacteria will collapse. Nevertheless, in our work, after operation, the bacteria still maintain their cell structure only with specific electroporation pores (~100 nm) on the surface. The bacterial morphology confirmed little contribution of Joule heating to microbial inactivation. We have added the discussion in the revised manuscript.

<Line 386>

“In addition, considering the relatively long contact time (from several seconds to minutes) needed for thermal disinfection, the Joule heating isn't likely to contribute to microbial inactivation due to the short contact time (0.025 s)^{15, 42}.”

<Added reference>

[15] Wang D, et al. Iron oxide nanowire-based filter for inactivation of airborne bacteria. *Environ. Sci. Nano* **5**, 1096-1106 (2018).

[42] Jin CY, Li Z, Williams RS, Lee KC, Park I. Localized temperature and chemical reaction control in nanoscale space by nanowire array. *Nano Lett.* **11**, 4818-4825 (2011).

<Line 399>

“Commonly, after disinfection treatment based on the mechanisms of chemical oxidation, charge transfer, ion toxicity, Joule heating, and/or mechanical damage, the bacteria will collapse due to the damaged outer structure^{41, 43-46}. Nevertheless, after RV-disinfection, the cell structure still maintained only with specific electroporation pores (~100 nm) on the bacterial surface. This could be additional strong evidence confirming the disinfection mechanism to be electroporation.”

<Added reference>

[41] Wang G, et al. An antibacterial platform based on capacitive carbon-doped TiO₂ nanotubes after direct or alternating current charging. *Nat. Commun.* **9**, 1-12 (2018).

[43] Valiei A, et al. Hydrophilic mechano-bactericidal nanopillars require external forces to rapidly kill bacteria. *Nano Lett.* **20**, 5720-5727 (2020).

[44] Vecitis CD, Zodrow KR, Kang S, Elimelech M. Electronic-structure-dependent bacterial cytotoxicity of single-walled carbon nanotubes. *ACS Nano* **4**, 5471-5479 (2010).

[45] Chen W, et al. Silver nanowire-modified filter with controllable silver ion release for point-of-use disinfection. *Environ. Sci. Technol.* **53**, 7504-7512 (2019).

[46] Li J, et al. Zinc-doped Prussian blue enhances photothermal clearance of *Staphylococcus aureus* and promotes tissue repair in infected wounds. *Nat. Commun.* **10**, 1-15 (2019).

Mechanical damage

When bacteria attach to some nanomaterial with sharp structures such as nanowires, nanosheets, and nanopillars, the membrane can decompose due to mechanical damage. However, to achieve microbial inactivation based on mechanical damage, normally, a relatively long contact time from several seconds to minutes is necessary⁴³. Due to the short contact time in our work (0.025 s), the mechanical damage isn't likely contributing a lot to disinfection. Furthermore, if bacteria were inactivated by mechanical damage, the cell membrane will collapse. While in our work, after operation, only small electroporation pores formed on the bacterial outer structure with a relatively complete cell structure. This also confirms the disinfection mechanism in the present work is electroporation rather than mechanical damage. To make our statement clear, we have added more discussion and references in the revised manuscript.

[43] Valiei A, et al. Hydrophilic mechano-bactericidal nanopillars require external forces to rapidly kill bacteria. *Nano Lett.* **20**, 5720-5727 (2020).

<Line 388>

“When microbes were attached to the nanowire-modified electrodes, the sharp structure of the nanowire may cause mechanical damage to the microbes and contribute to disinfection. However, to achieve microbial inactivation based on mechanical damage, a relatively long contact time from several seconds to minutes is commonly necessary^{43, 44}. Due to the short contact time in our work (< 0.025 s), the mechanical damage may only have little contribution to microbial inactivation.”

<Added reference>

[43] Valiei A, et al. Hydrophilic mechano-bactericidal nanopillars require external forces to rapidly kill bacteria. *Nano Lett.* **20**, 5720-5727 (2020).

[44] Vecitis CD, Zodrow KR, Kang S, Elimelech M. Electronic-structure-dependent bacterial cytotoxicity of single-walled carbon nanotubes. *ACS Nano* **4**, 5471-5479 (2010).

Comment 2:

Figure S8 shows the flow controller at the inlet and the flow sensor behind the electrodes. If the written flow rates in this manuscript are set values of the flow controller at the inlet, the actual flow near the electrodes should be much slower because the cross-section of the inlet looks much smaller than that of the duct. The actual flow rates near the positive electrode in the duct should be used to discuss the speed of inactivation.

Response 2:

Thanks for the reviewer’s helpful comment. The authors feel that our statement in our manuscript is not clear. In the experiment, all the flow rates were set according to the measured values of the flow sensor behind the electrodes in the duct. The so-called “flow controller” in Supplementary Fig. 8 is a pressure reducing valve of the compressed gas tank only showing the value of air pressure instead of flow speed. To make our statement clear, we have revised the Supplementary Fig. 8 as follows:

Supplementary Fig. 8 Structure of the V-TENG-powered RV-disinfection system.

Comment 3: Simply mentioning COVID-19 and building ventilation systems would be fine as potential applications in the discussion part, but not in the abstract or first paragraph of the introduction as it can mislead readers. There is not enough connection between this study and the specific examples.

Response 3:

We have revised the related statement in the abstract and the first paragraph of the introduction in the revised manuscript as follows.

<Abstract>

“Air-transmitted pathogens may cause severe epidemics (e.g., ~~COVID-19~~) showing huge threats to public health.”

<Line 39>

“Air-transmitted pathogens are the primary reason for people catching pneumonia, asthma, and influenza, ~~as well as the recent COVID-19 virus~~, as such, they remain a huge threat to public health.”

<Line 88>

“Our work successfully provides a proof-of-concept to confirm the application potential of this method for air disinfection in the ventilator systems of buildings in actual situation.”

<Line 92>

“In our work, we aim to provide a proof-of-concept to confirm the feasibility of RV air disinfection methods in the ventilator systems of buildings during the normal ventilation process. ~~The RV air disinfection method is designed to be deployed in the ducts of indoor buildings to achieve air disinfection during the normal ventilation process.~~”

Comment 4:

$-\log(C/C_0)$ is used as y-axes to show the efficacy of the system as the values are drastically different as shown in Figures 3, 4, and 5. However, it disturbs the quantitative analysis; for example, <38% inactivation is still a high value, but <0.21 log seems very low (row 220). Please consider changing the y-axes as inactivation % as needed.

Response 4:

Thanks for the reviewer’s valuable comment. Due to the relatively high microbial concentration and the high requirement of disinfection performance, sometimes even > 90% of microbial inactivation efficiency is not enough. Even one live pathogen that survives from disinfection may cause disease. The log removal efficiency ($-\log(C/C_0)$) is thus commonly used in the research of disinfection to quantify the disinfection efficiency of those more than 90%^{17, 28}. For example, the treatment time or chemical input dosage to achieve a 3-log (or 4-log) of microbial removal efficiency is commonly used to evaluate the disinfection performance for a specific disinfection technology⁴⁹. In addition, in some research, although the disinfection efficiency was plotted in percentage (%; C/C_0), the data shown in y-axes was still in the logarithmic form (e.g., 10^{-1} , 10^{-2} , 10^{-3} , and 10^{-4})³⁷. To make our statement clear, we deleted the expression of percentage and added more discussion and reference in the revised manuscript.

[17] Liu C, *et al.* Conducting nanosponge electroporation for affordable and high-efficiency disinfection of bacteria and viruses in water. *Nano Lett.* **13**, 4288-4293 (2013).

[28] Ding W, *et al.* TriboPump: a low-cost, hand-powered water disinfection system. *Adv. Energy Mater.* **9**, 1901320 (2019).

[49] Boudaud N, *et al.* Removal of MS2, Q β and GA bacteriophages during drinking water treatment at pilot scale. *Water Res.* **46**, 2651-2664 (2012).

[37] Liu C, *et al.* Rapid water disinfection using vertically aligned MoS₂ nanofilms and visible light. *Nat. Nanotechnol.* **11**, 1098-1104 (2016).

<Line 218>

“While almost no E. coli were inactivated (< 0.21 log removal efficiency ~~corresponding to <38% inactivation~~) when the output power of the V-TENG was disconnected.”

<Line 289>

However, with the no-charge-model RV-disinfection, only ~0.5 log removal efficiency (~~68% microbial inactivation~~) was achieved under the same operating conditions.

<Line 350>

However, when using the Cu₃PNP-Cu electrode, the RV-disinfection system was ineffective on the tested model microbes (< 0.15 log removal efficiency ~~corresponding to <29% inactivation~~) under the same operating conditions.

<Line 489>

“The log removal efficiency can easily express the level of inactivated microbes with high concentration and quantify the disinfection efficiency of those more than 90%⁴⁹.”

<Added reference>

[49] Boudaud N, et al. Removal of MS2, Q β and GA bacteriophages during drinking water treatment at pilot scale. *Water Res.* **46**, 2651-2664 (2012).

Comment 5:

Reference needs to be added on row 135.

Response 5:

We have added the reference on the row 135 in the revised manuscript as follows.

<Added reference>

[20] Huo Z-Y, et al. A Cu₃P nanowire enabling high-efficiency, reliable, and energy-efficient low-voltage electroporation-inactivation of pathogens in water. *J. Mater. Chem. A* **6**, 18813-18820 (2018).

[32] Zhang Z, Dua R, Zhang L, Zhu H, Zhang H, Wang P. Carbon-layer-protected cuprous oxide nanowire arrays for efficient water reduction. *ACS Nano* **7**, 1709-1717 (2013).

Reviewer # 2

In this work, authors have reported a rapid disinfection method for inactivating air-transmitted bacteria and viruses using the nanowire-enhanced localized electric field to damage the outer structures of microbes using the triboelectric nanogenerator that converts mechanical vibration to electricity for the air disinfection purpose. These are some questions that the authors need to clarify.

Comment 1:

Authors claimed that they have reported first self-powered disinfection system. However, this type of system is already reported: *Nano Energy* 2017, 36, 241-249 (A self-powered sterilization system with both instant and sustainable anti-bacterial ability) and *Nano Lett.* 2014, 14, 10, 5603–5608 (Static Electricity Powered Copper Oxide Nanowire Microbicidal Electroporation for Water Disinfection).

Response 1:

The authors appreciate the reviewer's helpful comment. To our knowledge, our work was the first one to report the self-powered disinfection system for air disinfection. The references (*i.e.*, Nano Energy 2017, 36, 241-249 and Nano Lett. 2014, 14, 10, 5603–5608) mentioned in this comment are focusing on water disinfection. To make a clear statement, we deleted the expression of “first” in the revised manuscript as follows.

<Line 81>

“Here, we report a ~~the first~~ self-powered disinfection system for the rapid disinfection of air-transmitted bacteria and viruses based on a highly efficient nanowire-assisted electroporation mechanism powered by vibration-driven TENGs (V-TENGs) that harvest mechanical vibration energy.”

Comment 2:

Formation of Cu₃P nanowires and nanoparticles should be confirmed from XRD and XPS analysis.

Response 2:

The authors appreciate the reviewer's helpful suggestion. We have added the measurement of XRD and XPS of Cu₃P nanowires and nanoparticles to confirm the material components. We have added a new discussion in the revised manuscript and new data in Supplementary Fig. 16.

<Line 345>

“Formations of Cu₃P for both NWs and NPs samples were confirmed from X-ray diffraction (XRD) and X-ray photoelectron spectroscopy (XPS) analysis (Supplementary Fig. 16).”

<Line 445>

“The material components were analyzed by XRD (PANalytical, Alpha 1 MPD). The chemical compositions were analyzed by an XPS using an Axis Ultra instrument (Kratos Analytical, K-alpha) with ultrahigh vacuum (<10⁻⁸ Torr) and a monochromatic Al K α X-ray source.”

<Supplementary Information>

Supplementary Fig. 16 X-ray photoelectron spectroscopy (XPS) and X-ray diffraction (XRD) analysis of Cu₃P nanowires and nanoparticles-modified electrode confirming formations of Cu₃P for both NWs and NPs samples. (a and b) Cu 2p spectra of the Cu₃PNWs (a) and Cu₃PNPs (b) -modified copper electrodes. The major peaks shown in both Cu₃PNWs and Cu₃PNPs samples at 932.9 eV for the Cu 2p_{3/2} energy level are attributed to Cu^{δ+} in Cu₃P¹. (c) XRD pattern of the Cu₃PNPs-modified copper electrode. Besides the two strong diffraction peaks at 43.4° and 50.6° from the Cu substrate (JCPDS file No. 04-0836), the other peaks can be assigned to the Cu₃P phase. The diffraction pattern for the Cu₃PNPs-modified copper electrode exhibits six peaks at 36.0°, 39.1°, 41.6°, 45.1°, 46.2°, and 47.3°, corresponding to (112), (202), (211), (300), (113), and (212) of Cu₃P phase (JCPDS file No. 71-2261), respectively².

<Added reference in Supplementary Information >

(Ref. SI) Fan et al. "Half-cell and full-cell applications of highly stable and binder-free sodium ion batteries based on Cu₃P nanowire anodes." *Advanced Functional Materials* 26.28 (2016): 5019-5027.

(Ref. SI) Huo et al. "A Cu₃P nanowire enabling high-efficiency, reliable, and energy-efficient low-voltage electroporation-inactivation of pathogens in water." *Journal of Materials Chemistry A* 6.39 (2018): 18813-18820.

Comment 3:

In the entire manuscript it is not clear which experiments are conducted in nebulizer and which one in ventilator? Were they conducted in same setup or with different apparatus? Authors should clarify those issues.

Response 3:

The authors appreciate the reviewer's helpful comment. In our work, all the experiments were performed in a duct connected by a nebulizer to generate bioaerosols and control humidity. We aim to provide a proof-of-concept to confirm the feasibility of this method for air disinfection in the ventilator systems of buildings in actual situations. To make this statement clear, we have added more statements and discussion in the revised manuscript.

<Line 88>

“Our work successfully provides a proof-of-concept to confirm the application potential of this method for air disinfection in the ventilator systems of buildings in actual situation.”

<Line 92>

“In our work, we aim to provide a proof-of-concept to confirm the feasibility of RV air disinfection methods in the ventilator systems of buildings during the normal ventilation process. ~~The RV air disinfection method is designed to be deployed in the ducts of indoor buildings to achieve air disinfection during the normal ventilation process.~~”

<Line 151>

“During operation, the TENG is placed on a shaker with aimed frequency and amplitude to mimic the actual operating conditions of the ventilator in a building.”

<Line 194>

“The disinfection performance of the V-TENG-powered RV-disinfection method for air-transmitted microbes was evaluated using a prototype to simulate the actual applications (Supplementary Fig. 8). In our work, feed solutions containing a high concentration of bacteria or viruses were added into a super-fine air compressed nebulizer and bacterial or viral bioaerosols were generated by the nebulizer to flow through the duct.”

Comment 4:

In the Figure 3g, why the live bacterial concentration changes periodically or more specifically decrease when the TENG is off? What is the reason of reducing the bacterial concentration even if electric field is not applied?

Response 4:

The authors deeply appreciate the reviewer’s valuable comment. Considering the air-transmitted microbes are commonly in an unstable and changeable concentration in the actual situation, the disinfection performance of our RV-disinfection method for treating intermittent (or in pulse shape) microbial bioaerosols in the airflow was evaluated. The nebulizer with the feed solution of microbes was turned on for 5 min and then off for 5 min to generate an intermittent (or in pulse shape) microbial bioaerosol. In Fig. 3g, “**start**” means turning on the nebulizer, and “**stop**” means turning off the nebulizer. The nebulizer was turned on and off for several cycles, and we only mark on the figure once to avoid repeated statements. The periodically turning on and off the nebulizer leads to a periodic change of the bacterial concentration. Thus, the reason for reducing the bacterial concentration when TENG was not applied is because that the nebulizer was turned on and off periodically. When turning off the nebulizer, the bacterial concentration will reduce. To make our statement clear, we added more discussion in the revised manuscript as follows.

<Line 243>

*“...the nebulizer with the feed solution of bacteria (*E. coli*) or viruses (MS2) was turned on (start in Fig. 3g) for 5 min and then off (stop in Fig. 3g) for 5 min to generate an intermittent microbial bioaerosol. This periodically turning on and off the nebulizer leads to a periodic change of the microbial concentration.”*

<Caption of Fig 3g>

“...The nebulizer with the feed solution was turned on (start) and then off (stop) for 5 min to generate an intermittent microbial bioaerosol and this periodically turning on and off leads to a periodic change of the microbial concentration.”

Comment 5:

If the distance between the positive and negative electrode is kept constant in no charge model (similar to the distance between positive and ground electrode) and one of the electrodes is surface modified with nanostructure, it is obvious that local electric field enhancement will also be there. Then why the antibacterial efficiency is poor in case of no charge model? Author should explain this clearly.

Response 5:

This is a very good question. The enhanced localized electric field exists near the surface of the nanowire-modified electrode in the no-charge-model system. However, without the charging process (in the no-charge-model), only trace surface charge will exist on the microbes. When flowing between the positive and negative electrodes in the no-charge-model system, due to the fast airflow rate (2 m/s) and short travel time (<0.025 s), microbes with trace surface charge (<10⁻¹⁵ C) cannot be trapped on the positive electrode surface because of the weak electrostatic attraction³⁹. Because the enhanced localized electric field only exists near the surface of the electrode, electroporation disinfection only occurs when microbes approach the electrode surface. Thus, the antibacterial efficiency is poor in the no-charge-model system. The significant difference in the disinfection efficiency between the charge and no-charge-model system also confirms the important role of microbial charging. Only after contacting with the macro-mesh electrode the microbes can be negatively charged (*E. coli* cell or MS2 particle carried charges ~ 6.1×10⁻¹⁰ or 7.2×10⁻¹² C, respectively; Fig. 4e). Then these charged microbes can be trapped by the positive electrode to achieve high-performance disinfection. To make our statement clear, we added more discussion in the revised manuscript as follows.

[39] Mainelis, G., *et al.* Electrical charges on airborne microorganisms. *J. Aerosol Sci.* (2001).

<Line 290>

“Without the charging process, only trace charge exists on the microbes. Although the enhanced localized electric field exists near the surface of the nanowire-modified positive

electrode in the no-charge-model, these microbes cannot approach the positive electrode surface due to the weak electrostatic attraction and short traveling time (< 0.025 s). Because electroporation disinfection only occurs when microbes approach the electrode surface, the antibacterial efficiency is poor in the no-charge-model system.”

Comment 6:

Authors should present the simulation of electric field distribution for Cu_3P NPs also in order to confirm the level of field enhancement due to these nanostructures.

Response 6:

The authors appreciate the reviewer’s helpful suggestion. We have added the simulation of electric field distribution near Cu_3P NPs (50 and 100 nm). Simulation results confirmed that NPs had the relatively weak ability to enhance the electric field. We have added more discussion in the revised manuscript and new data in Supplementary Fig. 17 as follows.

<Line 356>

“When applied with similar operating conditions, the electric field near Cu_3PNPs is insufficient for disinfection according to the simulation results (Supplementary Fig. 17).”

<Supplementary Information>

Supplementary Fig. 17 Simulation of electric field distribution near the surface of Cu_3PNPs (diameter of 100 and 50 nm) driven by the V-TENG, showing the relatively weak enhancement of the localized electric field. The scale bar is 500 nm.

Comment 7:

Author should check the TENG assisted antibacterial performance in different range of humidity (30% to 80%) as in real environment humidity varies a lot.

Response 7:

The authors appreciate the reviewer’s helpful suggestion. We have added the test to measure the disinfection performance under different humidity and added more discussion in the revised manuscript and new data in Supplementary Fig. 14.

<Line 256>

*“Furthermore, the disinfection performance under various humidity was evaluated. All the tested bacteria (*E. coli*) and viruses (MS2) can be completely inactivated in a wide range of humidity (from 30% to 80%) at a fast airflow rate (2 m/s; Supplementary Fig. 14).”*

<Supplementary Information>

Supplementary Fig. 14 Disinfection performance at various humidity. Feed solutions containing a high concentration of bacteria (*E. coli*) or viruses (MS2) were added into a super-fine air compressed nebulizer and bacterial or viral bioaerosols were generated by the nebulizer to flow through the duct. The initial concentration of MS2 is relatively higher than that of *E. coli*. Thus, when the microbes are completely inactivated, the log removal efficiency of MS2 (> 4.4-log) and *E. coli* (> 4.0-log) will be different. The airflow rate in the duct was fixed at 2 m/s using compressed gas. The humidity was controlled from 30% to 80% using another nebulizer to generate water aerosols in the duct. Dashed lines indicate that all microbes are inactivated, and no live microbes can be detected. All the tested bacteria (*E. coli*) and viruses (MS2) can be inactivated completely in a wide range of humidity (from 30% to 80%) at a fast airflow rate.

Comment 8:

In page 5, line 92-93, the authors mentioned that “The RV air disinfection method designed to be deployed in the ducts of indoor buildings to achieve air disinfection during the normal ventilation process.” Therefore, the authors should demonstrate the real time application of RV disinfection process embedding the device in the duct of domestic environment.

Response 8:

The authors deeply appreciate the reviewer’s valuable comment. In our work, we aim to provide a proof-of-concept to confirm the application potential of this method for air disinfection in the ventilator systems of buildings in actual situations. To make this statement clear without any misunderstanding, we have added a sentence and revised discussion in the revised manuscript.

<Line 88>

“Our work successfully provides a proof-of-concept to confirm the application potential of this method for air disinfection in the ventilator systems of buildings in actual situation.”

<Line 92>

“In our work, we aim to provide a proof-of-concept to confirm the feasibility of RV air disinfection methods in the ventilator systems of buildings during the normal ventilation process. ~~The RV air disinfection method is designed to be deployed in the ducts of indoor buildings to achieve air disinfection during the normal ventilation process.~~”

Comment 9:

The mechanism of bacterial destruction is not clear in the manuscript. Authors confirmed that chemical oxidation and intercellular ROS generation were not taking place. Then the authors should specifically mention the cause/mechanism responsible for killing bacteria in RV disinfection system.

Response 9:

The authors have added more statements in the revised manuscript to emphasize the mechanism responsible for killing bacteria and viruses in the RV disinfection system.

<Line 408>

“In summary, based on the above mechanism investigation, the cause responsible for microbial inactivation in our RV disinfection system is confirmed to be electroporation.”

Comment 10:

Since the filtered air will go to the environment, authors should also provide the biocompatibility tests of their materials as Cu₃P toxicity is a well-known phenomenon.

Response 10:

The authors appreciate the reviewer’s helpful suggestion. We have added the biocompatibility tests of the released Cu₃P. We collected 1 m³ of air after flowing through the disinfection device in 500 mL of sterilized DI water. Bacterial solution (1 mL; 10⁷ CFU/mL) was fed in the collecting water and cultivated at a fixed temperature (25 °C) for 48 h. During cultivation, the concentration of the live bacteria was investigated. To make a comparison, another 1 mL of bacteria solution (10⁷ CFU/mL) was added in DI water to investigate the concentration of the live bacteria with the same operating condition as control. During the cultivation process, the concentration of live bacteria in both collecting water with Cu₃P and DI water (control sample) was similar and didn’t decrease in 48 h. This confirmed that the released Cu₃P was ineffective for bacterial inactivation. The toxicity of the collecting water is low due to the low

concentration of the released Cu_3P . We have added this new experiment in Supplementary Fig. 18 and added new discussion in the revised manuscript as follows.

<Line 380>

“In addition, the released Cu_3P was confirmed to be ineffective for bacterial inactivation due to its low concentration based on the biocompatibility test (Supplementary Fig. 18).”

<Supplementary Information>

Supplementary Fig. 18 *Biocompatibility tests of the released Cu_3P . Air (1 m^3) after flowing through the disinfection device was collected in sterilized DI water (500 mL). Bacterial solution (1 mL; 10^7 CFU/mL) was fed in the collecting water and cultivated at a fixed temperature ($25 \text{ }^\circ\text{C}$) for 48 h. During the cultivation process, the concentration of the live bacteria was investigated. To make a comparison, another 1 mL of bacteria solution (10^7 CFU/mL) was added in DI water to investigate the concentration of the live bacteria with the same operating condition. During the cultivation process, the concentration of live bacteria in both collecting water with Cu_3P and DI water (control sample) was similar and didn't decrease. This confirmed that the released Cu_3P was ineffective for bacterial inactivation. The toxicity of the collecting water is low due to the low concentration of the released Cu_3P .*

REVIEWERS' COMMENTS

Reviewer #1 (Remarks to the Author):

The authors addressed well all the critics that this reviewer raised. Therefore, the current version is ready to be published.

Reviewer #2 (Remarks to the Author):

I do appreciate that the authors made lots of efforts in order to improve the quality of this manuscript. Belows are my comments regarding the revision of this manuscript:

1. Response to comment 1 is justified and necessary changes made, are acceptable.
2. Formation of Cu₃P nanowires and nanoparticles from XRD and XPS and their analysis provided are correct.
3. Experimental queries are clear from the response to comment 3.
4. The added discussion makes the comment 4 clear.
5. Explanation about poor antibacterial efficiency in case of no charge model is clear.
6. Satisfied with the simulation results and discussion.
7. Satisfied with the provided data of antibacterial performance in different range of humidity (30% to 80%).
8. Changes made are acceptable.
9. The proposed mechanism is clear.
10. Satisfied with the biocompatibility tests to check toxicity.